

# Bicycle balance assist system reduces roll and steering motion for young and older bicyclists during real-life safety challenges

Leila Alizadehsaravi and Jason K. Moore

Biomechatronic and Human-Machine Control Section, Biomechanical Engineering Department, Faculty of Mechanical, Maritime, and Materials Engineering, Delft University of Technology, Delft, South Holland, The Netherlands

## ABSTRACT

Bicycles are more difficult to control at low speeds due to the vehicle's unstable low-speed dynamics. This issue might be exacerbated by factors such as aging, disturbances, and multi-tasking. To address this issue, we developed a prototype 'balance assist system' with Royal Dutch Gazelle and Bosch eBike Systems at Delft University of Technology, which includes an electric motor capable of providing additional steering torque. We implemented a speed-adaptive feedback controller to generate the additional steering torque to that of the rider. We conducted a study with 18 older and 14 younger cyclists to first examine the effect of aging, disturbances, and multi-tasking on cycling at lower forward speeds, and evaluate the effectiveness of the system in improving the stability of the rider-bicycle system while facing these challenges. The study consisted of two scenarios: a single-task scenario where participants rode the bicycle on a marked narrow straight-line track, and a multi-task scenario where participants performed a shoulder check task and followed visual cues while tracking the straight-line. We introduced handlebar disturbances using the steer motor in half of the trials in both scenarios. All trials were repeated with and without the balance assist system. We calculated the bicycle mean magnitude of roll and steering rate—as indicators of bicycle balance control and required steering actions, respectively—and the rider's mean magnitude of lean rate with respect to the ground to investigate the effect of the balance assist system on rider's lateral motion. Our results showed that aging, disturbances, and multi-tasking increased the roll rate, and the balance assist system was able to significantly reduce it. The effect size of the balance assist system in reducing the roll rate across all conditions was found to be larger in older cyclists, indicating a more substantial impact compared to younger cyclists. Disturbances and multi-tasking increased the steering rate, which was successfully reduced by the balance assist system. Aging did not significantly affect the steering rate. The rider's lean rate was not significantly affected by age, disturbances, or the balance assist, indicating that the upper body plays a minor role when riders have good steering control authority. Overall, our findings suggest that lateral motion and required steering action can be affected by age, multi-tasking, and handlebar disturbances which can endanger cyclists' safety, and the balance assist system has the potential to improve cycling safety and reduce the incidence of single-actor crashes. Further investigation on riders' contribution to control actions is required.

Corresponding author
Leila Alizadehsaravi,
l.alizadehsaravi@tudelft.nl

## INTRODUCTION

Cycling is an eco-friendly means of transport that enhances healthy lifestyles and is favored by many people. Over the past decade, there was an increasing societal interest in electric bicycles (e-bikes) where the number of e-bikes sold in Europe increased from 0.5 million in 2009 to 5 million in 2021 (*News, 2020*). E-bikes enable riders to cycle for longer duration and distance by reducing physical fatigue (*Hoj et al., 2018*). However, with increased numbers of e-bikes, bicycle accidents due to inadequate steering and balance control by older cyclists have increased (*Lefarth et al., 2021*; *Berk et al., 2022*).

Bicycles are statically unstable but under certain conditions, *e.g.*, higher forward speeds and rider's control, can become stable (*Astrom, Klein & Lennartsson, 2005*). In The Netherlands in 2020, 70% of cycling crashes were reported to be single-actor with slippery surfaces and loss of balance as the main causes (*Krul et al., 2022*). Bicycle balance control requires a mixture of passive (bicycle's self-stability) and active (rider) control to ride the bicycle in a stable balanced state such that the rider-bicycle system could quickly restore if subjected to disturbances. Various factors, including tire characteristics, road unevenness, and wind disturbances, can influence bicycle dynamics (*Dell'Orto, Ballo & Mastinu, 2022*; *Schwab, Dialynas & Happee, 2018*). Thus, maintaining balance, pedaling, and steering require continuous physical and cognitive effort, and external disturbances create additional challenges for riders (*SWOV, 2017*; *Schwab, Dialynas & Happee, 2018*; *Afschrift et al., 2022*).

Accurate bicycle control relies significantly on factors such as steering angle and torque feedback. Visuo-vestibular and proprioceptive feedback has proven crucial for stabilizing the bicycle, particularly in relation to roll rate (*Dialynas et al., 2023*). Due to the degradation in sensory and motor organs, balance control in older adults is poorer than in young adults (*Alizadehsaravi et al., 2020*; *Afschrift et al., 2022*). Furthermore, aging compromises the ability to effectively distribute attention across multiple domains during multitasking (*Malcolm et al., 2015*). In the context of low-speed and multi-task cycling, older cyclists exhibit higher mean magnitude roll rate and mean absolute steering angle, indicating age-related challenges in controlling the inherent instability of bicycle motion (*Kovácsová et al., 2016*).

Apart from the effects of aging, differences in riding skills, contribute to distinct bicycle postural control strategies. These strategies may also vary in response to both internal and external disturbances. Skilled riders, regardless of the speed, exhibited less variability in both steering input and lean angle compared to novices when riding an instrumented bicycle on a roller trainer (*Cain, Ashton-Miller & Perkins, 2016*). Enhancing balance control could be beneficial for improving safety among older adults, unskilled cyclists, and regular cyclists facing challenging situations.

After the first prototype presented in *Nieuwenhuizen & Schwab (2017)*, we developed a second prototype balance assist bicycle together with Royal Dutch Gazelle and Bosch eBike Systems aiming to enhance safety by facilitating easier bicycle balancing, thus minimizing the risk of instability. We hypothesized that external disturbances and multi-tasking affect the bicycle motion and that our balance assist system improves the stable damped response in steer and roll rate. We also hypothesized that older cyclists have less lateral control authority than younger cyclists and the balance assist system is more effective for older participants. Reduction in roll rate (lateral motion) and steering rate variables are expected based on the reduced demand for compensatory and acute steering control, respectively, as the balance assist system applies enough control input to maintain or regain the bicycle's balance.

## METHODS

The aim of the study was to evaluate the effects of the balance assist system in situations where cycling is challenging, especially for older cyclists.

### Design and implementation of the speed-adaptive feedback controller

The dynamics of the bicycle are represented by a pair of coupled second-order differential equations (*Papadopoulos, 1987*): $M \begin{bmatrix} \ddot{\phi} \\ \ddot{\delta} \end{bmatrix} + v C_1 \begin{bmatrix} \dot{\phi} \\ \dot{\delta} \end{bmatrix} + (v^2 K_2 + g K_0) \begin{bmatrix} \phi \\ \delta \end{bmatrix} = \begin{bmatrix} T_\phi \\ T_\delta \end{bmatrix}$ where $(\phi, \delta)$ are time-dependent quantities, representing the roll and steer angles, and $(T_\phi, T_\delta)$ represent the applied roll and steer torques $(T_\phi, T_\delta)$, respectively. The constant matrices $M$, $C_1$, $K_0$, and $K_2$ represent various properties of the bicycle, and $v$ and $g$ represents the forward speed and acceleration due to gravity, respectively. We evaluated the stability of the uncontrolled bicycle by computing the eigenvalues of the system, derived from the characteristic equation of the benchmark bicycle (*Meijaard et al., 2007*) illustrated in the top panel of the Fig. 1: $\det(M \lambda^2 + v C_1 \lambda + g K_0 + v^2 K_2) = 0$.

$$M = \begin{bmatrix} 80.817 & 2.319 \\ 2.319 & 0.298 \end{bmatrix}, C_1 = \begin{bmatrix} 0 & 33.866 \\ -0.850 & 1.685 \end{bmatrix} \tag{1}$$

$$K_0 = \begin{bmatrix} -80.95 & -2.600 \\ -2.600 & -0.803 \end{bmatrix}, K_2 = \begin{bmatrix} 0 & 76.597 \\ 0 & 2.654 \end{bmatrix} \tag{2}$$

The real part of a complex eigenvalue, represented as $a$ in $\lambda = a \pm bi$, represents the rate of exponential decay or growth in the system's response (this can be derived from $e^{at}$). In the interpretation of our eigenvalues graph, a positive real part implies an unstable mode and the imaginary part $b$ gives the oscillation frequency, which is expressed as $2\pi f$, where $f$ is the frequency in Hz $(1/s)$.

In the uncontrolled bicycle, the weave mode (shown with a yellow o-shaped marker line in Fig. 1) becomes unstable at low forward speeds (top left panel; 1 to 4.3 m/s), while the capsize mode (depicted with a solid blue line in Fig. 1) becomes unstable at higher forward speeds (top middle panel; 4.3 to 6 m/s). The system's eigenvalues revealed that the uncontrolled bicycle, at forward speeds between 1 and 10 m/s, is laterally unstable, apart

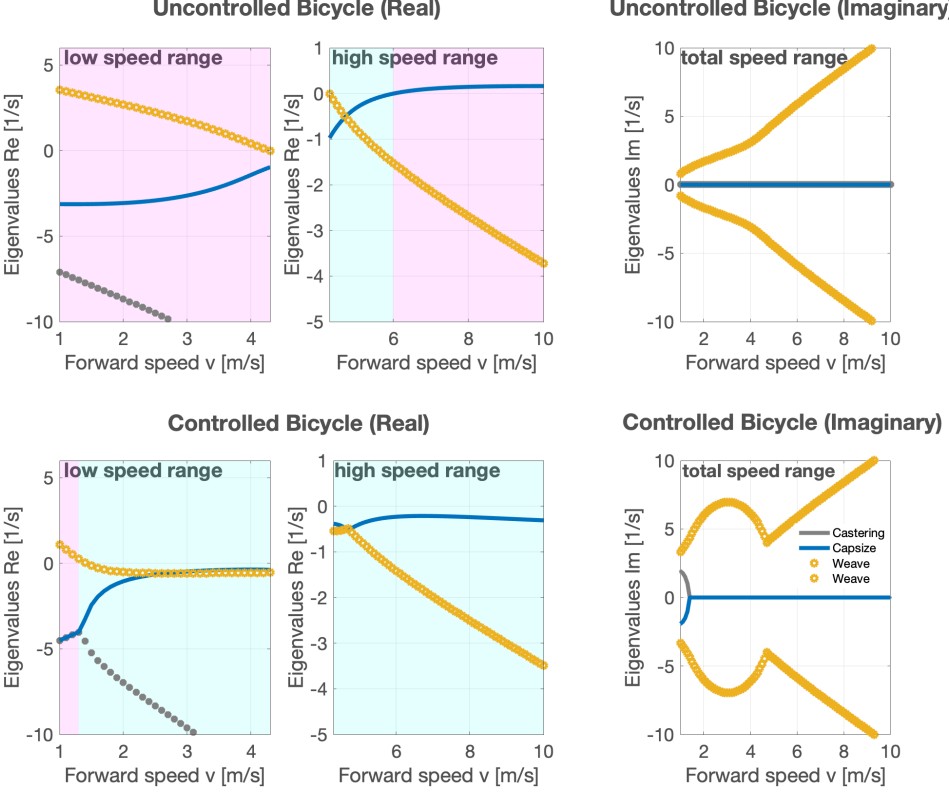

**Figure 1** The figure illustrates the stability and behavior analysis of a bicycle with a rigid rider under varying forward speeds, with and without the speed-adaptive feedback controller. Negative real parts indicate stability, while positive real parts indicate instability. The imaginary part represents the system's oscillatory behavior, with the magnitude of the imaginary part corresponding to the frequency of oscillation. The legend identifies the motion modes associated with the eigenvalues: 'Castering', 'Weave', and 'Capsize'. Unstable and stable forward speed regions are highlighted in pink and cyan within the figure, respectively.

from a range approximately between 4.3 to 6 m/s, where it exhibits lateral self-stability (*Meijaard et al., 2007*).

Subsequently, the speed-adaptive feedback controller was incorporated into the model. Our control objective was to stabilize the system at all forward speeds, specifically within the linearized dynamics about an upright position, with a zero-steer angle, and a constant forward speed. Our system is represented by the linear state-space model $\dot{x} = Ax + Bu$, where $\mathbf{x} = [\phi, \delta, \dot{\phi}, \dot{\delta}]^T$ is the state vector ($\dot{\phi}$ and $\dot{\delta}$ are representing roll rate and steering rate, respectively), and $u = [T_\delta]$ is the control input vector. The coefficient matrices $A$ and $B$ are given by:

$$A = \begin{bmatrix} 0 & I \\ -M^{-1}(K_0 + v^2 K_2) & -M^{-1} v C_1 \end{bmatrix}, B = \begin{bmatrix} 0 \\ M^{-1}[0,1]^T \end{bmatrix} \tag{3}$$

The control input $T_\delta$ adapts to feedback from the roll rate and roll angle, the predominance of each being dependent on the forward speed (*Schwab, Kooijman & Meijaard, 2008*). At speeds less than 4.7 m/s, $T_\delta$ responds only to roll rate feedback,

whereas it relies more on roll angle feedback at speeds exceeding 4.7 m/s. Both types of feedback are modulated by a speed-dependent control gain.

At low speed, the control gain, $K_L$, is defined by the equation $K_L = -K_v(v_{max} - v_n)$, where $v_n$ represents the current speed, $v_{max}$ denotes the critical speed (this is the speed at which the bicycle transitions between two unstable regions, influenced by different modes based on eigenvalues; it can be chosen to be between 4.3 to 6 for the bicycle presented), and $K_v$, ranging from 6 to 14, is a constant for lower speeds. Thus, the control input is given by $u = T_\delta = K_L\dot{\phi}$.

For high-speed situations, the control gain, $K_H$, is given by $K_H = -K_c(v_n - v_{max})$, where $K_c$ is a constant for higher speeds, set at 0.8 (*Schwab, Kooijman & Meijaard, 2008*). This formula yields the control input as $T_\delta = K_H\phi$.

We then manually tuned $K_v$ and $K_c$ to maximize stability across most of the speed range while simultaneously minimizing the required steering torque. The stability as indicated by the negative real part eigenvalues was achieved by iteratively tweaking the feedback gains. Our results demonstrated that with the implementation of this controller, the stability region of the bicycle shifted to lower forward speeds (1.3 m/s) and, at higher speeds, the bicycle motion was marginally stable compared to its uncontrolled motion (Fig. 1).

It should be noted that the model, which does not account for the rider's relative motion to the bicycle, might yield slightly different results in comparison to real scenarios. Therefore, the controller in the bicycle has some proprietary elements that improve behavior in edge cases. To validate our findings, we conducted subjective pilot tests with riders using the chosen gains before the experiment. The resulting motion was perceived as smooth and responsive by the riders, without any signs of excessive steering, thereby validating our approach. We kept the control gain constant throughout the experiment for all participants.

## Instruments

The participants rode a Gazelle step-through city electric bicycle (Arroyo C8 electric), that has been modified as a balance assist bicycle with a custom direct drive steering motor embedded in the headtube (Fig. 2). This motor is meant to provide an additional torque between the headtube and the steer tube, helping the rider in the balancing task. We instructed all participants to use the same constant gear setting while cycling. They were asked to maintain a self-selected 'constant' speed between 2 and 5 m/s in eco mode (the mode with the lowest propulsion assist). This approach aimed to leverage the bicycle's instability at lower speeds. To verify that participants adhered to the speed instructions, we conducted a post-hoc ANOVA analysis (see Appendix S1, Speed).

With balance assist prototype, the bicycle's forward speed, roll and steer angle and rate are measured or estimated. The balance assist bicycle is equipped with a wheel speed sensor on the rear wheel. Additionally, a Bosch steering angle sensor is positioned at the steering tube to measure the steering angle and steering rate. This sensor has an absolute physical resolution of 0.1 degree and operates at a sampling rate of 100 Hz. Furthermore, an inertial measurement unit (IMU) is mounted on the rear frame, allowing for accurate measurement of the roll rate and estimation of the roll angle. Additionally, it includes data

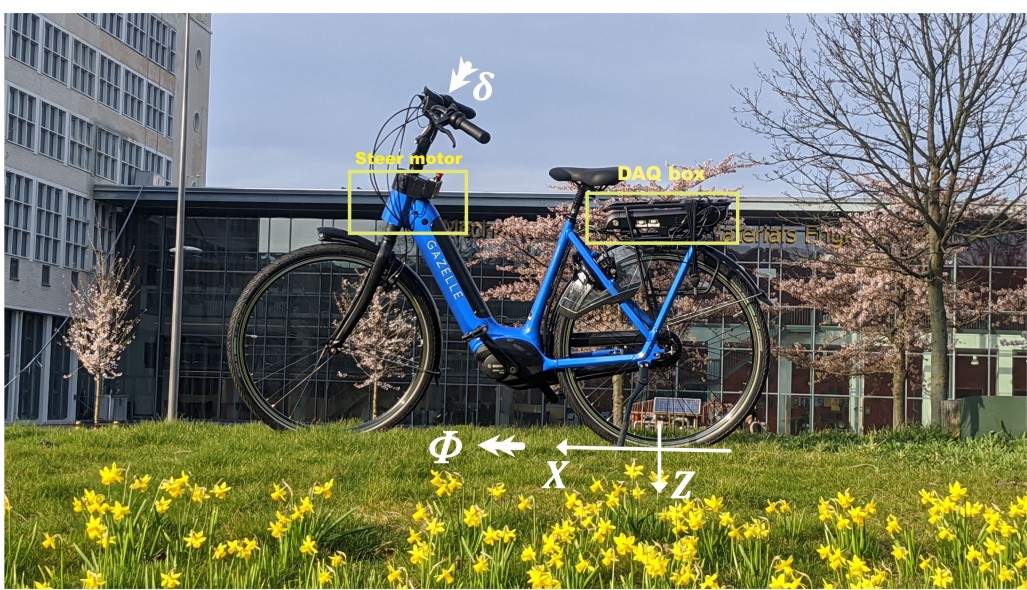

**Figure 2 Second prototype balance assist bicycle at Delft University of Technology in collaboration with Royal Dutch Gazelle and Bosch eBike Systems.** The steer motor and the data acquisition box (DAQ) are annotated in the image. The roll angle ($\phi$) around the $X$ axis (forward) and the steering angle ($\delta$) around the steer axis are also shown.

acquisition and control boards located in its rear luggage rack. The steer motor applies up to 7 Nm torque between the rear frame of the bicycle and the steering tube. We evaluated the effectiveness of the balance assist system in young and old cyclists by creating a series of conditions comparable with real-life cycling challenges. To directly investigate the rider's motion, a 3-axis IMU sensor (Shimmer Research Ltd, Dublin, Ireland) was mounted on the spine at the T7 level by an elastic strap on the back of the participant (Appendix S1, Fig. 1) prior to performing the experiment.

## Participants

Thirty-two participants (18 old, 67 ± 4 years old, 174 ± 8 cm, 83 ± 14 kg, 6 females; and 14 young, 23 ± 2 years old, 179 ± 10 cm, 71 ± 12 kg, three females) participated in this study. The participants were recruited by advertisements. All participants signed a written consent form and were able to cycle and had no balance disorders or history of injury or fall caused by instability over the last year. The Human Research Ethics Committee of the Delft University of Technology (The Netherlands) approved the experiments (Letter of Approval 2080).

## Experimental procedure

All participants first cycled for 2–5 min to get familiarized with the bicycle with and without the balance assist system (blind setup), and to reduce the habituation effect throughout the experiment. Then participants performed 16 trials divided in two scenarios (single- and multi-task cycling; Fig. 3).
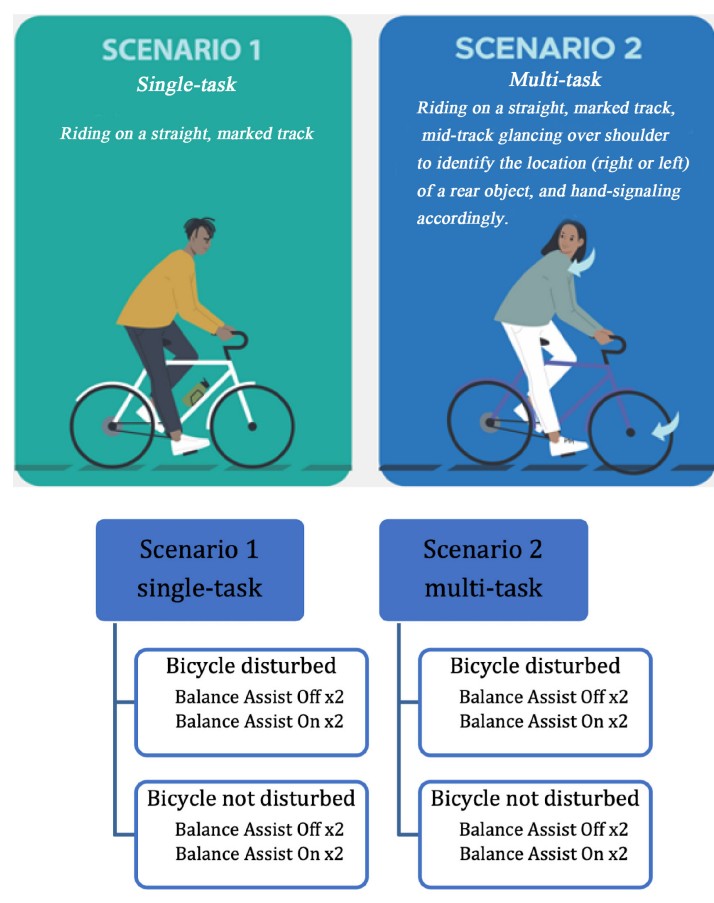

**Figure 3** **Experimental procedure shows in total sixteen trials in two scenarios, four conditions per scenario with repetition; both scenarios were performed with and without the balance assist system.** First scenario, single-task in presence and absence of disturbances (0.5 s), and second scenario, multi-task with shoulder check in presence and absence of long disturbances (1 s).

To minimize potential biases, we employed a randomized sequence for the trials. However, we consistently scheduled single-task scenarios before multi-task scenarios. This approach was intentionally chosen to mitigate situations where an older participant might be unable to complete all trials, ensuring the collection of a complete dataset for at least one scenario. This precaution proved crucial during our study when one participant was unable to complete the full range of trials.

To maintain consistency in posture throughout all trials, participants were instructed to sit upright with both hands on the handlebars. We individually adjusted the saddle position for each participant to ensure a consistent and upright seating position.

### Single- and multi-task scenarios

A common and simple task often experienced during natural cycling is tracking a constant heading without deviating laterally from a straight path, for example when you ride along a straight cycle path alongside a fellow cyclist, along a narrow cycle path, or close to a cycle path edge. To mimic this task, we asked our participants to ride at a constant self-selected

low speed (2 to 5 m/s), along a 30 (m) straight line highlighted with a 5 (cm) width road-tape on the ground.

A common multi-task scenario was simulated by asking the participants to do a shoulder check task and follow instruction corresponding to the identified visual cues while tracking the above-mentioned line. The instruction was to look back at the starting point over their preferred shoulder the moment they reached the red cone in the middle of the track. At the starting point the researcher was holding up a cone randomly in her left or right hand. Participants were instructed to identify that direction and lift/place back their corresponding hand off/on the handlebar while following the track as closely as possible. The hand opposite to the identified direction always remained on the handlebar.

Each scenario comprises four conditions, each repeated once, resulting in a total of eight trials (2 Balance Assist System states (on/off) × 2 Disturbance states (on/off) × 2 repetitions) per scenario (Fig. 3). Data for the single-task scenario is available for all participants. However, four participants (three older and one younger) have missing data in the multi-task scenario; three due to system malfunction in hot weather conditions under the high load of self-induced perturbation using the steer motor, and one older participant was unable to complete the multi-task scenario. These four participants were therefore excluded from the statistical analysis for the multi-task scenario. For the rider's upper body lean rate, data was available for 24 participants, including six younger and 18 older participants.

### Disturbances

In half of the trials (in total and per scenario) participants were subjected to small disturbances induced by the steering motor when the balance assist system was(de)activated (Appendix S1, Fig. 2). The purpose was to evaluate the bicycle behavior when an unwanted disturbance applies to the bicycle and causes difficulty in control, such as when you hit a bump in the road (short-duration) or front rack cargo or the wind gust pulls the steer in an undesired direction (long-duration). The disturbances were implemented by 3 square wave pulses with random intervals resulting in ∼1.2 Nm steering torque. This perturbation is significant because steer torques during straight-line riding tasks are generally less than 5 Nm (*Moore, 2012*). The disturbances began one second after the forward speed reached 2 m/s for the first time in that trial. The duration of the disturbances were 0.5 s and 1 s in single-task and multi-task cycling, respectively.

## Data analysis

We collected time series data from various sensors, including the steering angle sensor, the rear frame IMU, and the IMU placed on the rider's torso. We analyzed the data to obtain the bicycle's steering rate, roll rate, and rider's upper body lean rate relative to the ground as the outcome measures (dependent variables).

To study the effect of different conditions on these dependent variables, we first extracted and segmented relevant data from the sensors, along with forward speed and yaw rate data. We manually identified the start of cycling when the forward speed increased from 0 m/s. We then divided the time series data into two phases: the transient phase when accelerating

from zero to a steady-state phase where the speed was approximately constant until the end of the track. For analysis, we used the steady-state phase of the time series data as a segment of interest (Appendix S1, Fig. 3).

To determine the start indices for each segment of interest, we used the first index at which forward speed reached 1.5 m/s in the accelerating phase. To determine the end indices for each segment of interest, we used two criteria: the first index at which either forward speed reached 1.5 m/s in the decelerating phase or the absolute difference in yaw rate was greater than or equal to 4, indicating a change in cycling direction at the end of the cycling track. We chose 4 deg/s yaw rate as a threshold for starting to turn after visually inspecting all trials.

We updated the start and end time points to segment all signals from the bicycle's and rider's IMUs and the steering angle sensors. To reduce high-frequency noise in the signals, we applied a low-pass second-order Butterworth filter with a cutoff frequency of 25 Hz on the roll rate and steering rate signals. We also detrended those resulting filtered signals by subtracting their mean values.

We then calculated the mean absolute steering rate, roll rate, and lean rate of the time-series data over each segment that represented an approximate steady-state traversal. Finally, we calculated the average of two repetitions per condition to reduce the effect of randomness in balance control.

Note that, desirably, we wanted two repetitions per condition. However, in a few cases, due to malfunctions, the system did not apply the perturbation, resulting in trials being labeled as not disturbed. For the analysis, we reported the average of the maximum number of repetitions per condition. If subjects had no trials performed for a particular condition, we excluded them from the analysis. For illustration purposes, we included representative time series of bicycle motion data of all conditions in single-task scenario in Fig. 4.

### Bicycle's steering motion; steering rate

We quantified the steering rate by the mean absolute steering rate in deg/s. The steering rate is the rate of change of steering angle around the steer axis $\delta$ (Fig. 2). Higher steering rate correlates to the total (rider +motor) steering control effort to stabilize the vehicle. Since we have not measured the rider's steering effort separately, the results here are a summation of the rider and the steering motor's contribution to steering motion. We assumed that less steering rate corresponds to a reduced combined effort of the rider and motor, in those cases when the balance assist system is activated. If the steering rate is higher in the condition where the balance assist system is activated compared to when it is not, we cannot draw a strong conclusion on the rider's effort in motion control.

### Bicycle's lateral motion; roll rate

We quantified the bicycle's lateral balance control by the mean absolute roll rate of the bicycle. The roll rate is the rate of change of angle of bicycle rear frame around the $X$ (or forward) axis (Fig. 2). A lower roll rate indicates that the system is better damped around the vertical axis when the system is subjected to internal (noise in rider's motor control) or external disturbances (steering motor).
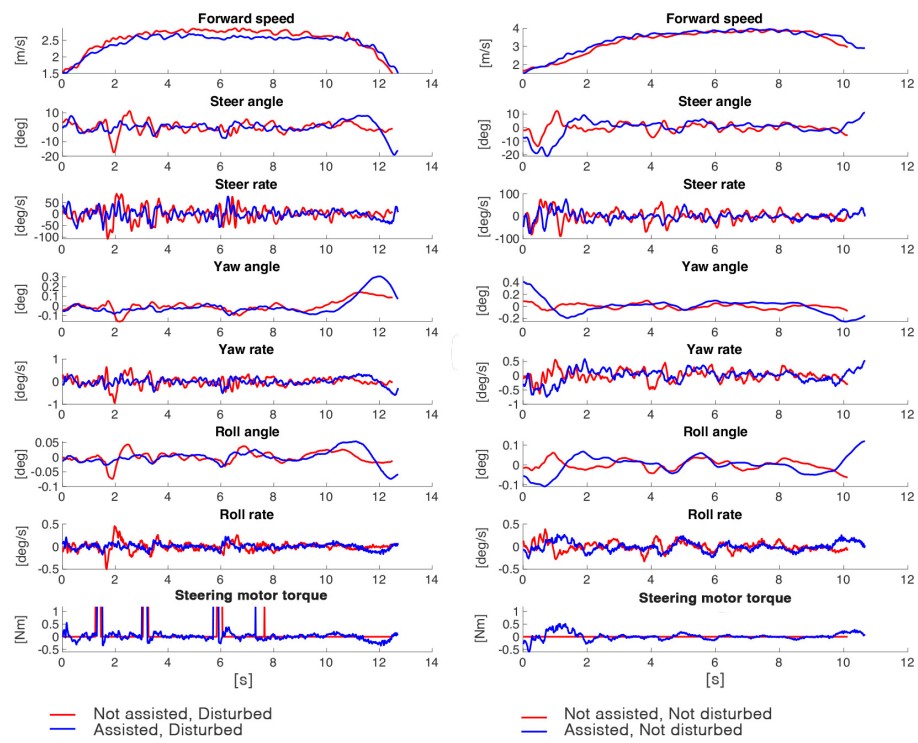

**Figure 4   Bicycle motion data for one representative participant in single-task scenario for all conditions.** The *x* axis shows the number of samples.

### Rider's lateral motion; lean rate

The IMU's *Y* axis was aligned with the participant's spine, *X* axis was vertical to *Y* in the frontal plane, and *Z* axis was vertical to *Y* axis in the sagittal plane (Appendix S1, Fig. 1). We evaluated the rider's postural balance control by the lean rate defined as the rate of change of torso angle in Shimmer IMU *XY* plane or around the *Z* axis relative to the ground. The lean rate indicates how close the rider maintains to the zero-configuration state.

## Statistics

To evaluate the effects of the Balance Assist System, Disturbances, and Aging on dependent variables (steering, roll, and rider's lean rate), we performed repeated measures ANOVA with Balance Assist System (on/off) and Disturbances (on/off) as within-subject factors and Aging as a between-subjects factor. In the case of a significant main effect, we investigated the interactions of effects, and if significant, post-hoc ANOVA was performed to test the effect of Balance Assist System on affected variables.

To evaluate the effects of Scenarios (multi-tasking) we performed the repeated measures ANOVA on dependent variables with Balance Assist System (on/off) and Scenarios (single-task/multi-task) as within-subjects, and Aging as a between-subjects factors on Undisturbed trials. Since the duration of disturbances was different in single- and multi-task cycling, to

**Table 1** The effects of balance assist system, disturbances, and aging on bicycle roll and steering and on rider's lean rate in single-task cycling. Bold values indicate statistically significant effects.

| Scenario 1; single-task | Steering rate | | Roll rate | | Lean rate | |
|---|---|---|---|---|---|---|
| | F(1,30) | p | F(1,30) | p | F(1,17) | p |
| Balance assist | **39.273** | **<.001** | **47.235** | **<.001** | 1.344 | 0.262 |
| Disturbances | **31.198** | **<.001** | **11.573** | **0.002** | 0.509 | 0.485 |
| Age | 2.04 | 0.163 | **6.675** | **0.015** | 2.739 | 0.116 |
| Balance assist * Age | 0.967 | 0.333 | 1.58 | 0.218 | 0.108 | 0.747 |
| Disturbances * Age | 1.606 | 0.205 | 1.398 | 0.246 | 0.142 | 0.711 |
| Balance assist * Disturbances | 0.348 | 0.560 | 0.853 | 0.363 | 0.117 | 0.737 |
| Balance assist * Disturbances * Age | 1.406 | 0.245 | 0.37 | 0.548 | 0.055 | 0.818 |

evaluate the effect of Scenarios, the disturbed trials were excluded for a fair comparison between two scenarios.

Finally, in case of effectiveness of the balance assist system in improving the outcome measures, to gain insights into how the balance assist system affects each age group separately, we conducted a simple main effects analysis. We divided the data by age group and performed an ANOVA on each group to examine the effect of the balance assist system on the roll rate and steering rate.

We performed the statistical analysis in JASP (University of Amsterdam, The Netherlands) version 0.16, and $p < 0.05$ was considered significant.

# RESULTS

The results in Tables 1 and 2, show the effects of Balance Assist System, Disturbances and Aging on bicycle steering rate, roll rate, and the rider's lean rate. Table 3 shows the effect of Balance Assist System, Scenarios (multi-tasking) and Aging on bicycle steering rate, roll rate, and the rider's lean rate for undisturbed conditions.

## Effects of balance assist system, disturbances, and age per scenario
### Effects of the balance assist system, disturbances, and age in single-task cycling

In single-task cycling, a significant effect of the Balance Assist System and Disturbances was found on the steering rate, without any interaction between main factors. The results indicate that the steering rate was higher in disturbed compared to undisturbed cycling ($t = 5.586, p < 0.001$, Cohen's $d = 0.987$), and that activation of the Balance Assist System, regardless of age or disturbances, significantly reduced steering rate ($t = -6.267, p < 0.001$, Cohen's $d = -1.108$; Fig. 5, Table 1).

Regarding bicycle roll rate in single-task cycling, significant effects of the Balance Assist System, Disturbances, and Age without any interactions between main effects were observed, (Fig. 5, Table 1). Post-hoc analysis showed that roll rate was higher in disturbed compared to undisturbed cycling, and in older compared to younger adults ($t = 3.402$, $p = 0.002$, Cohen's $d = 0.601$; and $t = 2.584, p = 0.015$, Cohen's $d = 0.457$, respectively).

**Table 2  The effects of balance assist system, disturbances, and aging on bicycle roll and steering and on rider's lean rate in multi-task cycling.** Bold values indicate statistically significant effects.

| Scenario 2; multi-task | Steering rate | | Roll rate | | Lean rate | |
|---|---|---|---|---|---|---|
| | $F_{(1,26)}$ | $p$ | $F_{(1,26)}$ | $p$ | $F_{(1,17)}$ | $p$ |
| Balance assist | **10.556** | **0.003** | **21.377** | **<.001** | 0.293 | 0.595 |
| Disturbances | **111.635** | **<.001** | **60.394** | **<.001** | 2.152 | 0.161 |
| Age | 3.208 | 0.085 | 3.487 | 0.073 | 3.020 | 0.100 |
| Balance assist * Age | 0.975 | 0.333 | 0.03 | 0.864 | 0.189 | 0.669 |
| Disturbances * Age | 0.534 | 0.471 | 2.365 | 0.136 | $2e-6$ | 0.999 |
| Balance assist * Disturbances | 1.048 | 0.315 | 2.101 | 0.159 | 0.076 | 0.786 |
| Balance assist * Disturbances * Age | 2.281 | 0.143 | 1.904 | 0.179 | 0.076 | 0.786 |

**Table 3  The effects of balance assist system, scenarios, and aging on bicycle roll and steering and on rider's lean rate in undisturbed trials.** Bold values indicate statistically significant effects.

| Single- vs. Multi-task | Steering rate | | Roll rate | | Lean rate | |
|---|---|---|---|---|---|---|
| | $F_{(1,26)}$ | $p$ | $F_{(1,26)}$ | $p$ | $F_{(1,15)}$ | $p$ |
| Balance assist | **34.681** | **<.001** | **38.275** | **<.001** | 0.211 | 0.652 |
| Scenario | **19.678** | **<.001** | **14.041** | **<.001** | 0.514 | 0.484 |
| Age | 1.972 | 0.172 | **6.686** | **0.016** | 2.217 | 0.157 |
| Balance assist * Age | 0.064 | 0.803 | 1.634 | 0.212 | 0.248 | 0.626 |
| Scenario * Age | 2.054 | 0.164 | 0.414 | 0.525 | 0.239 | 0.632 |
| Balance assist * Scenario | 0.299 | 0.589 | 0.10 | 0.921 | 1.114 | 0.308 |
| Balance assist * Scenario * Age | 0.143 | 0.708 | 0.097 | 0.758 | 0.010 | 0.923 |

The balance assist system significantly reduced the roll rate ($t = -6.873$, $p < 0.001$, Cohen's $d = -1.215$).

In single-task cycling there was not a significant effect of Age or Disturbances nor Balance Assist System on rider's lean rate (Table 1).

### Effects of the balance assist system, disturbances, and age in multi-task cycling

In multi-task cycling, there was no significant effect of Aging on steering rate (Fig. 6, Table 2). However, there were effects of Balance Assist System and Disturbances on steering rate. While steering rate was higher in disturbed compared to undisturbed cycling ($t = 10.566$, $p < 0.001$, Cohen's $d = 1.997$), activation of the Balance Assist System, regardless of age or disturbances, significantly reduced steering rates compared to deactivation ($t = -3.249$, $p = 0.003$, Cohen's $d = -0.614$).

The analysis of roll rate in multi-task cycling revealed significant strong effects of Disturbances and the Balance Assist System on bicycle roll rate with no significant effects of Age or any interactions between the main factors (Fig. 6, Table 2). Post-hoc analysis showed higher bicycle roll rates in disturbed compared to undisturbed cycling ($t = 7.771$, $p < 0.001$, Cohen's $d = 1.469$). Activation of the Balance Assist System resulted in significantly lower bicycle roll rates, regardless of age or disturbances ($t = -4.623$, $p < 0.001$, Cohen's $d = -0.874$).

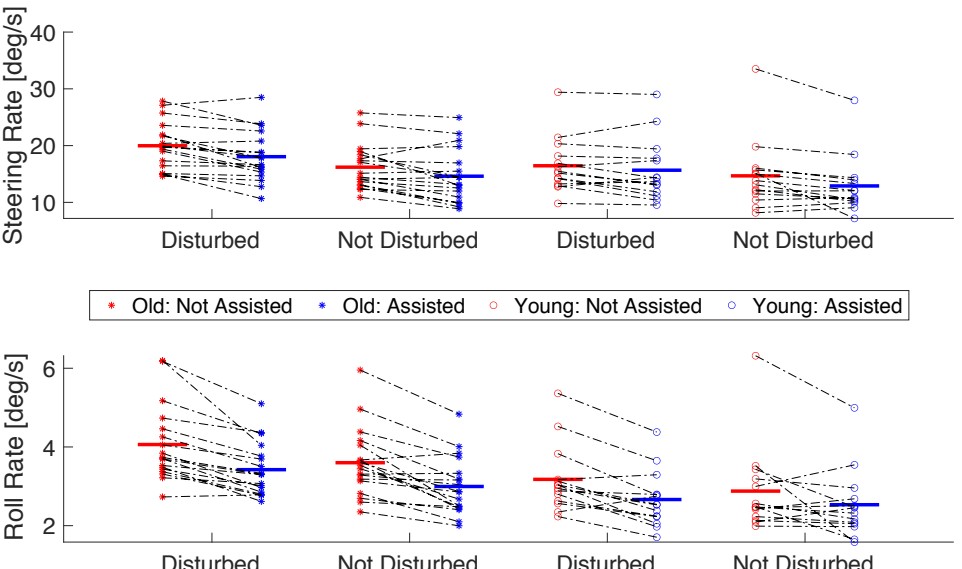

**Figure 5** **The effect of the Balance Assist System on roll and steering rate in single-task scenario.** Star-shaped and empty circle markers represent older and younger participants, respectively. Red indicates that the Balance Assist System is deactivated, while blue represents its activation. Black dot-dash lines connect individual participant data points, showing transition from the deactivated to the activated conditions. Horizontal lines in colors corresponding to the markers denote average values for all participants within a condition and an age group. Both disturbed and undisturbed conditions are presented.

In multi-task cycling, neither Age, Disturbances, nor the Balance Assist System had a significant effect on riders' lean rate (Table 2).

## Effects of balance assist system, scenarios, and age (undisturbed cycling)

There was no significant effect of Aging on the bicycle's steering rate. However, significant effects were observed for both Scenario and the Balance Assist System on the bicycle's steering rate (Table 3), without any interactions between these main factors. Post-hoc analysis showed that the steering rate during multi-task cycling was significantly higher than during single-task cycling ($t = 4.436$, $p < 0.001$, Cohen's $d = -0.838$. Additionally, activation of the balance assist system led to a decreased steering rate, irrespective of whether the cyclist was engaged in single- or multi-task cycling scenarios ($t = -5.889$, $p < 0.001$, Cohen's $d = -1.113$).

Furthermore, there were significant effects of Aging, Scenario, and the Balance Assist System on the bicycle's roll rate (Table 3). Post-hoc analysis showed that the roll rate was higher in older adults compared to young adults ($t = 2.586$, $p = 0.016$, Cohen's $d = 0.489$), and higher during multi-task scenarios compared to single-task scenarios ($t = 3.747$, $p < 0.001$, Cohen's $d = 0.708$). Regardless of Age and Scenario, the bicycle's roll rate was

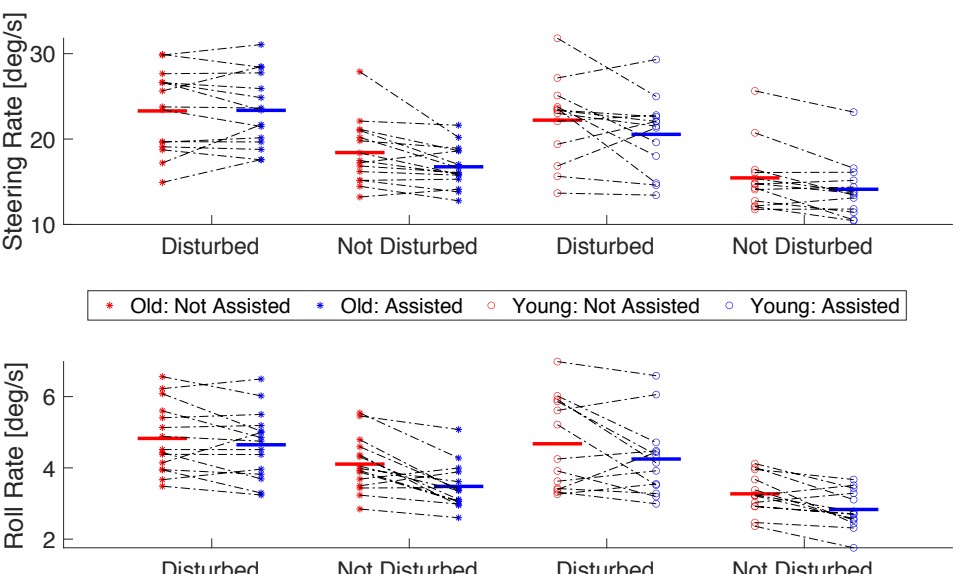

**Figure 6** **The effect of the Balance Assist System on roll and steering rate in multi-task scenario.** Star-shaped and empty circle markers represent older and younger participants, respectively. Red indicates that the Balance Assist System is deactivated, while blue represents its activation. Black dot-dash lines connect individual participant data points, showing transition from the deactivated to the activated conditions. Horizontal lines in colors corresponding to the markers denote average values for all participants within a condition and an age group. Both disturbed and undisturbed conditions are presented.

lower when the balance assist system was activated, compared to when it was deactivated ($t = -6.187$, $p < 0.001$, Cohen's $d = -1.169$).

There was not any effect of Aging, Scenario, or Balance Assist System on rider's lean rate (Table 3).

## DISCUSSION

At low forward speeds, a bicycle becomes very difficult to balance because the time-to-double of a bicycle's unstable motion can be as brief as 0.3 s (*Hess, Moore & Hubbard, 2012*). This difficulty is reflected in the very large motion variability observed during straight ahead cycling at low speeds (*Moore et al., 2010*). Applying external disturbances inevitably may cause growth in motion variability.

We aimed to reduce motion variability and thus rider control effort at low forward speed by introducing balance assisting steer torque. We expected reduced motion variability based on the reduced need for compensatory behavior (*Alizadehsaravi, 2021*) in the presence of assistive technology, as the bicycle states are known to the control unit. We designed and implemented a speed-adaptive feedback controller that laterally stabilizes the bicycle at low forward speeds thus rejecting small perturbations. We evaluated the system's effectiveness (regarding the lateral motion and required steering actions) in response to

real-life challenges experienced by the rider-bicycle system. We found that the balance assist system decreased the steering rate and reduced the lateral motion (roll rate) in all conditions in both age groups. This indicates that the bicycle's balance can be enhanced using the balance assist system. The rider's lean rate did not change with age, disturbances, or the balance assist system, and this could be caused by the limited role the upper body may play in bicycle balance control when both steer and lean are available as control inputs (*Sharp, 2008*).

Overall, our results suggest that a balance assist system can reduce unnecessary motions at low forward speed, during disturbances, and while multi-tasking. This system has the potential to increase safety in challenging balancing situations. We also conducted the statistical analyses using standard deviation instead of mean magnitude roll and steering rate, which produced comparable results (see Appendix S1).

## Balance assist system and aging

Our study found a significant effect of aging on roll rate and a trend towards higher steering rates among older participants, indicating reduced ability to control the bicycle's rolling motion and suggesting that age may impact bicycle balance control. Regardless of age, our results showed improved bicycle balance control when the balance assist system was activated.

Recent research suggests that aging impacts the brain's capacity for processing sensory inputs into motor actions (*Moulton, Rudie & Dukelow, 2022*), and our findings suggest that the balance assist system may assist older cyclists by processing the bicycle's states earlier than would occur in their central nervous system. This leads to a faster control action to damp lateral motion in the 'older rider-bicycle' system when the balance assist system is activated. Simple main effect tests showed that the balance assist system reduced roll rate in both single-task and multi-task cycling, with a greater effect on older cyclists' roll rate (single-task; Older: $F = 51.547$, $p < 0.001$, $\eta^2 = 0.752$, Young: $F = 10.386$, $p = 0.007$, $\eta^2 = 0.444$ and multi-task; Older: $F = 14.113$, $p = 0.002$, $\eta^2 = 0.502$, Young: $F = 8.355$, $p = 0.014$, $\eta^2 = 0.410$). Furthermore, e-bikes reduce the physical effort for propulsion, but do not provide fatigue reduction in the steering and balancing task. While steering in the long term may lead to fatigue and increase the risk of falling, especially in older cyclists (*Weavil et al., 2018*), the decreased steering rate using a balance assist system is promising to possibly reduce the fall risk. The balance assist system assisted older adults more than young in reducing their steering rate in single-task (single-task; Older: $F = 44.759$, $p < 0.001$, $\eta^2 = 0.116$, Young: $F = 8.675$, $p = 0.011$, $\eta^2 = 0.076$), and all undisturbed trials among both scenarios (Older: $F = 42.335$, $p < 0.001$, $\eta^2 = 0.751$; Young: $F = 9.226$, $p = 0.01$, $\eta^2 = 0.435$).

The increased number of single-actor bicycle crashes due inadequate balance control, especially in older cyclists, suggests that our system could induce balance control closer to young cyclists, potentially leading to safer cycling and reduce the number of age-related accidents. However, future research with larger sample sizes, various maneuvers, and a longer duration of cycling may be needed to confirm this trend and determine its significance.

### Balance assist system and disturbances

Internal and external disturbances are challenges that cyclists encounter during cycling due to internal sensory and motor organ noise or environmental factors. Results showed that the disturbances increased the roll rate and reduced the lateral balance control authority. We found a reduced roll rate, which indicates an enhanced rider's ability to control the bicycle's rolling motion, when using balance assist system.

In addition, disturbances also caused an increase in the steering rate, which was effectively reduced by the balance assist system. This suggests that the balance assist system aids in regaining balance control with fewer steering actions following disturbances. This could be beneficial for cyclists in the long term, as it can reduce physical fatigue. In addition, it may also foster a healthier lifestyle by allowing people to ride for longer distances. Our findings suggest that, on average, the use of the balance assist system led to a decrease in the amount of steering actions required to achieve the perceived zero-configuration state, regardless of age and disturbances.

### Balance assist system and multi-tasking

In multi-task cycling, the steering rate and roll rate significantly increased, reflecting the higher cognitive and motor control demand of the additional tasks. This is especially challenging for older cyclists with lower working memory. The balance assist system showed improvements in lateral balance control and required steering actions during multi-task cycling. These benefits were more pronounced for older cyclists, particularly in undisturbed trials, suggesting that the balance assist system effectively conserves cognitive resources otherwise utilized for bicycle control. However, a closer look at Fig. 6 in the context of combining multi-tasking and disturbances in older cyclists shows that the average mean magnitude of the steering rate among old participants with the activated balance assist system did not significantly change compared to when it was deactivated (23.361 *vs.* 23.293 deg/s). However, the standard deviation of the mean magnitude of the steering rate among old participants slightly decreased (4.258 assisted *vs.* 4.754 not assisted). This suggests the need for further investigation into riders' behaviors to understand the balance assist system's effects in these conditions and may indicate that the balance assist system improved the consistency of steering rates among older participants, despite the minimal difference in the mean values. Overall, our results imply that multi-tasking, a necessity in real-world cycling, leads to increased motion variability and a balance assist system can potentially improve the balance control and reduces the risk of undesired motion. Cyclists, especially older participants with lower cognitive and physical fatigue thresholds, could benefit from using the balance assist bicycle for a long distance and extended-duration riding.

## Balance assist system and cyclists' lateral balance control; lean rate

We did not observe any effect on the rider's lean rate with respect to the ground. This might be due to the rider's natural tendency not to lean when they can also steer to maintain the bicycle's balance, aligning with optimal control predictions (*Sharp, 2008*).

Additionally, prior observations (*Kooijman, Schwab & Moore, 2009*; *Moore, Kooijman & Schwab, 2011*) provide evidence that riders tend to keep their upper body relatively

inertially stationary during straight riding, even if the bicycle's roll motion has relatively larger variability. In essence, it seems easier for riders to allow the low inertia bicycle to roll than to move their bodies along with it for control purposes. This could explain the lack of lean rate changes we observe in this study.

Alternatively, ineffectiveness of the balance assist system on the lean rate (torso lateral motion) could be explained by the fact that the balance assist controller does not measure or estimate the rider's state. While the controller succeeded in keeping the bicycle in an upright position with less lateral oscillation in presence of the balance assist system, it is not clear whether the riders were able to accept the control action induced by the steering motor without over-reacting. Therefore, further investigation is necessary to understand the rider's perception and intention.

## Limitations and suggestions

There are some limitations in our study to note. We collected the older participants' data at a parking lot at the Den Haag Gazelle User Experience Center and the young participants at a parking lot at TU Delft with slightly different road surfaces (asphalt *vs.* cycling path bricks). However, by ensuring that all data per participant was collected consistently within the same environment, we mitigated the risk of skewed results.

The core findings of our study regarding the effectiveness of the balance assist system were drawn from the within-subject changes in variables (variables activated - variables deactivated). Therefore, these conclusions can be considered largely independent of the test location. However, it is crucial to note that the impact of aging on bicycle balance control may not be entirely independent of the test location. Factors such as differences in road surfaces may influence these findings. Therefore, future research could look into controlling these variables more effectively to fully comprehend the relationship between age and bicycle balance control.

Moreover, in human–robot interactions, 'trust' is seen as a key to improve the operations and a feeling of safety (*Goillau, 2003*). Therefore, it is expected that when adapting to new assistive technologies, the positive effect of the assistive system is more pronounced when riders trust the technology and yield to the generated steering torque by the steering motor. Since there is no direct measurement of the rider's effort, again we cannot draw a strong conclusion whether that was by choice or the rider's put enough effort in optimizing the lateral motion variability. However, we have asked riders to perform trials in a consistent manner; therefore variation in effort is not expected per participant, and the results are more likely due to the action of the balance assist system rather than changes in the riders' action.

We observed that five out of 18 older adults (all males) failed at least once in performing the first multi-task trial. It might be interesting to study the effect of gender on multi-task cycling to address adequate regulations for older male cyclists. In future studies, we recommend estimating the lateral position of the bicycle and calculating the deviation of the rider-bicycle from the straight-line, especially during multi-task cycling.

One of the common issues among our older participants was having a stiff neck, which would not allow them to look over the shoulder without using their torso. The problem

with that is a higher variation of motion while their range of motion is limited compared to young participants and that influences their motion coordination and lateral balance. A small side mirror added to older participants' bicycles could potentially help to eliminate some of the hazards, but careful adaptation is necessary to accurately identify the direction and distance of motion of other objects through the mirror.

## CONCLUSIONS

Aging, disturbances, and multi-tasking are real-life challenges, and our study demonstrated that they can negatively affect cycling lateral balance and steering motion. Our balance assist system was shown to be effective in improving lateral balance by reducing lateral motion (indicated by the reduced roll rate) and steering rate. These effects were observed not only in single-task cycling but also in multi-task cycling, across both age groups, in the presence and absence of the disturbances. The effects of the balance assist system were more pronounced among older cyclists. To further enhance the effectiveness of the balance assist system under conditions where multi-tasking is combined with disturbances, considering the addition of human motion detection and estimation to the control algorithm could be beneficial.

## ACKNOWLEDGEMENTS

The authors would like to thank the individuals who participated in the experiment. They would also like to thank Arend Schwab and Marco Reijne for their contribution to developing the balance assist system. Additionally, the authors extend their thanks to David Gabriel, Felix Dauer, Oliver Maier, Maarten Pelgrim, and Sierd Heida for their feedback and valuable contributions to the development of the balance assist system. Special appreciation is also due to Frans van der Helm for his feedback throughout the study.

### Funding

This study is funded by Dutch Research Council, Nederlandse Organisatie voor Wetenschappelijk Onderzoek (NWO), under the Citius Altius Sanius program and in collaboration with Bosch eBike Systems and Royal Dutch Gazelle. The funders had no role in study design, data collection and analysis, decision to publish, or preparation of the manuscript.

### Grant Disclosures

The following grant information was disclosed by the authors:
Dutch Research Council.
Nederlandse Organisatie voor Wetenschappelijk Onderzoek (NWO), under the Citius Altius Sanius program.
Bosch eBike Systems and Royal Dutch Gazelle.

## Competing Interests

The authors declare there are no competing interests.

## Author Contributions

- Leila Alizadehsaravi conceived and designed the experiments, performed the experiments, analyzed the data, prepared figures and/or tables, authored or reviewed drafts of the article, and approved the final draft.
- Jason K. Moore conceived and designed the experiments, authored or reviewed drafts of the article, and approved the final draft.

## Human Ethics

The following information was supplied relating to ethical approvals (i.e., approving body and any reference numbers):

The Human Research Ethics Committee of the Delft University of Technology (The Netherlands) approved the study (#2080).

## Data Availability

The raw data are available in the Supplemental Files.

## Supplemental Information

Supplemental information for this article can be found online at http://dx.doi.org/10.7717/peerj.16206#supplemental-information.

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
