# Peer review of "Bicycle balance assist system reduces roll and steering motion for young and older bicyclists during real-life safety challenges"

_PeerJ, doi:10.7717/peerj.16206_

## Round 0.1 · original submission · Major Revisions

Both reviewers found merit in your manuscript. Please address both reviewers concerns, particularly related to the methods, citations, and address the concern related to your conclusion about effort.

·

Basic reporting

1. The article is well written, but there are some minor grammar issues throughout.
2. The article could better use existing knowledge about human-bicycle dynamics by making comparisons to other results and using prior knowledge to better motivate data analysis decisions. For example, recent work by G Dialynas, AL Schwab, and colleagues contains many measurements of human and bicycle dynamics; other work by Cain et al. (PLoS One 2016) contains significant data about steering, roll, and rider lean dynamics.
3. I was not able to find a few of the references and a few of them need additional details in the citations.
a. Additional details needed for:
i. Moore 2012
ii. Schwab et al. 2008
iii. Schwab et al. 2018
iv. Nieuwenhizen and Schwab 2017
b. I was unable to find the following references, either by general search or by searching the theses collection at TU Delft, which are both critical for understanding the steer assist design and algorithms:
i. Schwab et al. 2008
ii. Nieuwenhizen and Schwab 2017
4. The structure of the article is professional.
5. Figure 2 is not very useful and does not add much additional information to the article. In contrast, many figures are cited in Appendix 1. I would like to see more of the information from Appendix 1 in the main article. In particular, Figure S2 and Figure S6 panels A and B are very helpful for understanding the data collection and dataset.
6. Consider condensing Figure 3 and Figure 4, perhaps using grouped bar graphs to have Disturbed and Not Disturbed results on a single plot.
7. Figure S3 y-axis should be labeled.
8. Figure S5 panels A and B: provide units for ‘SteerMotor’ and please use the same y-axis limits between the left and right panels.
9. The submission is self-contained with results relevant to the hypotheses.

Experimental design

1. Research is original and falls withing the aims and scope of the journal.
2. Research questions are well defined.
3. Some concerns about how the research was conducted:
a. I was surprised that speed wasn’t more closely controlled (only specified cycling within a range of 2-5 m/s). Cain et al. 2016 shows that steering rate can change quite markedly for some riders in this speed range (Fig 8 in Cain et al. 2016). I am curious if the authors have any sense as to the influence of speed on the measured kinematics.
b. Was the order of the trials randomized? This is not clear from the description of the protocol.
c. Similarly, was the pedaling cadence controlled? Hopefully the authors can appreciate that pedaling cadence can have an enormous effect on roll rate and steering rate (and not necessarily roll angle and steer angle).
d. Depending on the posture of the riders, the z-axis angular rate captured by the IMU may not be an accurate estimate of torso lean rate.
4. The methods are not described with sufficient detail and information to replicate or to allow the reader to understand how the research addresses existing knowledge gaps.
a. The description of the design and implementation of the controller is insufficient. I was unable to find either paper referenced regarding the previous design (Nieuwenhuizen and Schwab 2017) or the ‘steer-into-the-fall' (Schwab et al., 2008). While the basic concept of the controller is understood, the details about the exact design and tuning process are lacking, which prevents others from replicating the design and prevents others from identifying ways to improve bicycle balance assist systems. Details must be added to the manuscript regarding the controller.
b. The researchers chose mean absolute angular rates to quantify performance rather than the standard deviation. Can the authors provide their motivation for this choice?

Validity of the findings

1. The analysis of the data set was appropriate.
2. The processed data (outcome metrics for each trial) on which the conclusions are based are provided. All data files can be opened and appear complete.
3. Figures do not appear to be manipulated.
4. Most conclusions are supported by the data, but I take issue with making conclusions about rider effort when the authors did not directly measure effort – they have only measured kinematics (steer, roll, and lean rates), which may or may not have direct and clear relationships to effort. While no statements about effort appear in the conclusions section, they do appear in the Discussion and Abstract. Unless the authors have evidence that steer rate is strongly correlated to effort/steering torque (please provide a reference), statements about effort should be excluded from the manuscript.

Additional comments

1. Overall, I enjoyed reading this manuscript. However, I feel the impact and value of the results is limited by the limited description of the balance assist algorithm.
2. Design and interpretation of the controller: If I interpret this correctly, the controller is using a bicycle model that uses the benchmark bicycle parameters, correct? Is the controller adapted in any way to account for different bicycle geometries or riders?
3. Figure S4: Can you provide some additional details regarding the ‘Current command from Teensey to Steering motor’ (what is it and how does it relate to the experiment)? Are these data from a disturbed trial?
4. The main revisions I would like to see are as follows (as discussed above):
a. Provide additional details about the controller for the balance assist
b. Provide additional details for the citations that I was unable to locate
c. Do not make claims about effort unless you can provide convincing evidence that steer rate is highly correlated to steering torque
d. Provide additional details about experimental design (randomization of trials, cadence, rider posture)

Reviewer 2 ·

Basic reporting

no comments

Experimental design

no comments

Validity of the findings

no comments

Additional comments

This paper evaluates the effectiveness of a balance assist system in older and young adults while cycling a straight line (with or without a dual task and perturbations). This is a well written paper. I have several minor comments/suggestions that might improve the readability of the paper.

Minor comments:
1. Methods L 88 – 95: You mention that you manually tuned the gains, but it is unclear how this was done ? It understand that you select gains that result in a locally stable controller (negative real parts eigenvalues) but how can you minimize required steering torque ? Do you mean minimize the feedback gains (and hence minimize required steering torque). You can maybe make this more explicit for the reader
2. Methods L95: slightly different results are to be expected in real scenarios. I assume that you only expect slightly different results when the subjects are indeed moving on the bicycle but do not control the steering angle. I guess that you expect large difference when the subject is also controlling the steering angle. You might want to clarify this to the reader…
3. L77-95: maybe make it more explicit that you are talking about local stability of the system (around the state of upright position of the bike with zero velocities) and not stability in general.
4. Figure 1: it might be easier for the reader if you add legends for different colours (eigenvalue 1, 2, 3). The reader might also like a black line for the maximal value of the real parts of eigenvalues (you use this to determine if the system is locally stable) and maybe highlight locally unstable from 1 - +/- 4.5 m/s in the controller off condition.
5. L103-110: You did not explain how you measured the roll angle and rate at this point (I assume using an IMU on the frame, is this correct ?).
6. L111: simple equation for feedback laws + report feedback gains would be nice in the main manuscript.
7. Figure 2: you might want to illustrate roll axis in this figure ? (this might help some readers without a mechanics background)
8. Figure 3: I think that you can improve this figure. Given the relatively small number of subjects I think that you have to show the individual datapoints or visualise the distribution (using for example a violin plot). As the balance assist on-off is a within- subjects variable (repeated measures) you also might want to connect the datapoints per subjects with a line as you do the statics on the difference between assist on-off for each subject.
9. L364-366: You might want to rewrite this sentence. I understand that the effect of the balance assist system in general is independent on the test location as this is a within-subjects variable. However, this sentence now suggests that the conclusions about age are also independent of the test location (which is not true).

---

## Round 0.2 · accepted · Accept

Both reviewers found the current manuscript acceptable.

·

Basic reporting

No comment

Experimental design

No comment

Validity of the findings

No comment

Additional comments

I thank the authors for their thoughtful and thorough responses to my and the other reviewer's critiques, comments, and questions. I think the manuscript has been greatly improved.

Reviewer 2 ·

Basic reporting

no comment

Experimental design

no comment

Validity of the findings

no comment

Additional comments

Thank you for considering my comments and for making adjustments to the manuscript. I would like to offer one minor suggestion: for Figure 4, could you please consider adding an xlabel to the figure and using 'time' instead of 'frames' on the x-axis?

---

## Author Rebuttal · Round 0.2

# Rebuttal Letter

Leila Alizadehsaravi; Jason K. Moore

10.07.2023

Dear Editor and Reviewers,

Thank you for providing your valuable feedback on our paper. We appreciate the time and effort you have dedicated to reviewing our work and offering insightful suggestions for its improvement. We have carefully considered each of your comments and made the necessary revisions to our manuscript accordingly. In our revised manuscript, any removed words are crossed out with a single line and colored in red, while any added words are underlined with a squiggle and colored in blue. Please find our detailed responses below:

# Reviewer 1

## Basic reporting

**Comment 1:** The article is well written, but there are some minor grammar issues throughout.

Response: We have thoroughly reviewed the manuscript and corrected all grammar errors.

**Comment 2:** The article could better use existing knowledge about human-bicycle dynamics by making comparisons to other results and using prior knowledge to better motivate data analysis decisions. For example, recent work by G Dialynas, AL Schwab, and colleagues contains many measurements of human and bicycle dynamics; other work by Cain et al. (PLoS One 2016) contains significant data about steering, roll, and rider lean dynamics.

Response: We have expanded our literature review to better include the recent works by G Dialynas, AL Schwab, and colleagues, along with Cain et al. (Lines 57-69).

**Comment 3:** I was not able to find a few of the references and a few of them need additional details in the citations. a. Additional details needed for: i. Moore 2012 ii. Schwab et al. 2008 iii. Schwab et al. 2018 iv. Nieuwenhizen and Schwab 2017 b. I was unable to find the following references, either by general search or by searching the theses collection at TU Delft, which are both critical for understanding the steer assist design and algorithms: i. Schwab et al. 2008 ii. Nieuwenhuizen and Schwab 2017

Response: We apologize for the oversight regarding the details and accessibility of some references. We have now included additional details for the references requested:

- Moore, 2012: Davis Instrumented Bicycle — Human Control of a Bicycle: Jason K. Moore. PhD thesis, University of California, Davis, Mechanical and Aerospace Engineering, Davis, CA. http://moorepants.github.io/dissertation/davisbicycle.htmltorque-wrench-experiments.

- Schwab et al., 2008: Schwab, A. L., Kooijman, J. D. G., and Meijaard, J. P. (2008). Some recent developments in bicycle dy- namics and control.
  http://bicycle.tudelft.nl/schwab/Publications/SchwabKooijmanMeijaard2008.pdf.

- Schwab et al., 2018: Schwab, A. L., Dialynas, G., and Happee, R. (2018). Some effects of crosswind on the lateral dynamics of a bicycle. https://doi.org/10.3390/proceedings2060218

Thank you for pointing out the reference to the MSc thesis of Nieuwenhuizen. We understand your concerns about its accessibility. Currently, this thesis is under embargo, which explains the difficulty in locating it. It is important to note that while the thesis relates to the first prototype, our current work discusses the second prototype which, though built on similar foundational principles, has a distinct design and structure. Specifically, both prototypes are designed around the principle of using sensors to read the bicycle's states and a steer motor to augment the rider's torque. We hope this clarification aids in understanding the context and we appreciate your patience and understanding on this matter.

**Comment 4:** The structure of the article is professional.

Response: We appreciate the acknowledgement of the professional structure.

**Comment 5:** Figure 2 is not very useful and does not add much additional information to the article. In contrast, many figures are cited in Appendix 1. I would like to see more of the information from Appendix 1 in the main article. In particular, Figure S2 and Figure S6 panels A and B are very helpful for understanding the data collection and dataset.

Response: Based on your feedback, we have revised Figure 2 to incorporate the bicycle axes from Figure S6. Furthermore, Figures 3 and 4 have been added to the main manuscript to enhance understanding of the data collection and dataset.

**Comment 6:** Consider condensing Figure 3 and Figure 4, perhaps using grouped bar graphs to have Disturbed and Not Disturbed results on a single plot.

Response: We have revised Figure 3 and Figure 4 for clarity. In each figure, the results for both Disturbed and Not Disturbed are presented in a single subplot. One subplot is dedicated to steer rate, and the other to roll rate. Additionally, we included data points for each participant and connected the 'with and without balance assist' conditions to further highlight the effect of the balance assist system. This format is consistently applied to both single-task and multi-task scenarios.

**Comment 7:** Figure S3 y-axis should be labeled.

Response: Figure S3 has now been updated and 4 conditions are depicted in 4 subplots and the y-axis is labeled for better understanding.

**Comment 8:** Figure S5 panels A and B: provide units for 'SteerMotor' and please use the same y-axis limits between the left and right panels.

Response: Figure S5 is now updated and added to the main manuscript and units are added and the ylim for assisted and not assisted are comparable.

**Comment 9:** The submission is self-contained with results relevant to the hypotheses.

Response: We appreciate the acknowledgement of the relevance of the results to our hypotheses.

## Experimental design

**Comment 1:** Research is original and falls within the aims and scope of the journal.

**Comment 2:** Research questions are well defined.

**Comment 3a:** I was surprised that speed wasn't more closely controlled (only specified cycling within a range of 2-5 m/s). Cain et al. 2016 shows that steering rate can change quite markedly for some riders in this speed range (Fig 8 in Cain et al. 2016). I am curious if the authors have any sense as to the influence of speed on the measured kinematics.

Response: Thank you for your insightful comments and for pointing out the impact of speed on steering rate as demonstrated in the Cain et al. 2016 study. We recognize the importance of speed as a factor in cycling dynamics and agree that it could potentially introduce variations in riders' behavior.

However, in our study design, we consciously chose to give riders a range of speeds (2-5 m/s) to operate within, rather than enforcing a specific speed. This decision was based on our intent to minimize interference with riders' control actions, aiming for a more naturalistic riding experience. We encouraged riders to maintain a constant pace to the best of their ability.

We also focused more heavily on exploring the influence of aging, balance assist, disturbance, and multitasking on riders' behavior. By having each participant perform all trials in a similar manner, we sought to minimize the possible effects of speed variation on the cycling motion across trials.

Furthermore, we addressed the potential impact of speed on our results in Appendix 1, where we conducted a post-hoc ANOVA analysis to confirm that subjects adhered to the instructions and maintained their speed. As long as each subject traveled at the same speed in their own trials, we can make conclusions on the effect of our independent variables: disturbances, balance assist, and multitasking. The analysis demonstrated no significant differences in average, standard deviation, and range of speed (maximum-minimum speed) between trials **per subject** ($p > 0.05$), effectively confirming our constant speed assumption.

That being said, we do understand your concerns regarding speed variability and its possible impact on the measured kinematics. We will consider these points in future studies, exploring more controlled speed conditions to more fully understand its effect on cycling dynamics.

**Comment 3b:** Was the order of the trials randomized? This is not clear from the description of the protocol.

Response: We would like to confirm that indeed, the trials were randomized. However, we did maintain a specific order where single-task scenarios were always performed before multi-task scenarios.

This approach was employed to ensure that in instances where a participant, particularly those in the older age group, may be unable to continue the experiment, we would at least have one complete set of data collected for a single-task scenario. This approach proved to be prudent, as we did encounter a situation with one older participant who was only able to perform the single-task scenario trials, which were administered in a randomized order.

We value your attentive review and the chance to provide additional detail about our experiment's structure in the manuscript lines 159-163.

**Comment 3c:** Similarly, was the pedaling cadence controlled? Hopefully, the authors can appreciate that pedaling cadence can have an enormous effect on roll rate and steering rate (and not necessarily roll angle and steer angle).

Response: Thank you for your comment. We acknowledge that pedaling cadence can indeed have an impact on roll rate and steering rate, as you pointed out. In our study, our aim was to capture a more realistic cycling behavior at low speed, and allowing participants to pedal at their preferred cadence aligned with that objective. While we instructed them to keep the cadence similar across conditions, individual variations may have influenced the results. But each subject used the same gear

setting and thus we can use the same reasoning as our answer to the speed question above.

We performed a statistical analysis approach that compared each subject's performance within different conditions, rather than comparing between subjects. This allowed us to focus on the effects within individuals and control for individual differences to a certain extent. Considering the circumstances and the goals of our study, we believe that allowing naturalistic cycling behavior was a suitable choice.

**Comment 3d:** Depending on the posture of the riders, the z-axis angular rate captured by the IMU may not be an accurate estimate of torso lean rate.

Response: We appreciate the reviewer's concern regarding the variations in riders' postures. In our study, participants were specifically instructed to sit upright with both hands on the handlebars. Additionally, the saddle position was individually adjusted to ensure an upright sitting posture. If participants had only minimal forward posture angles, the torso lean rate gyro reading would be minimally affected. Our instructions were designed to reduce posture variations and ensure consistency across participants. We have now included this clarification in the methods section for better transparency (lines 164-166).

**Comment 4:** The methods are not described with sufficient detail and information to replicate or to allow the reader to understand how the research addresses existing knowledge gaps.

**Comment 4a:** The description of the design and implementation of the controller is insufficient. I was unable to find either paper referenced regarding the previous design (Nieuwenhuizen and Schwab 2017) or the 'steer-into-the-fall' (Schwab et al., 2008). While the basic concept of the controller is understood, the details about the exact design and tuning process are lacking, which prevents others from replicating the design and prevents others from identifying ways to improve bicycle balance assist systems. Details must be added to the manuscript regarding the controller.

Response: We apologize for the lack of detail in describing the design and implementation of the controller. We have now included additional information in the manuscript regarding the controller, including its basic concept, design, and tuning process (lines 84-127).

**Comment 4b:** The researchers chose mean absolute angular rates to quantify performance rather than the standard deviation. Can the authors provide their motivation for this choice?

Response: We appreciate your query. Our choice to use mean absolute angular rates was to gauge the overall intensity of rotational movements displayed by participants. This approach aligns with Kovacsova et al. (Kovacsova2016), who used mean absolute angular rates when assessing low-speed cycling performance among middle-aged and older individuals.
Given that our timeseries data oscillated around zero (resulting in an average close to zero), both the standard deviation and mean absolute can produce similar results. This relationship is further detailed at: https://stats.stackexchange.com/questions/81986/mean-absolute-deviation-vs-standard-deviation. To demonstrate the consistency of our results using standard deviation values with our original findings using mean absolute angular rates, we have included the results for the standard deviation of roll rate and steer rate during both single- and multi-task cycling in Appendix 1. These findings align closely with those derived from the mean absolute of the respective measures. We have also incorporated this observation into the discussion: 'We conducted the statistical analyses also using standard deviation values instead of mean magnitude, yielding comparable results (see Appendix 1)'.

## Validity of the findings

**Comment 1:** The analysis of the data set was appropriate.

**Comment 2:** The processed data (outcome metrics for each trial) on which the conclusions are based are provided. All data files can be opened and appear complete.

**Comment 3:** Figures do not appear to be manipulated.

**Comment 4:** Most conclusions are supported by the data, but I take issue with making conclusions about rider effort when the authors did not directly measure effort – they have only measured kinematics (steer, roll, and lean rates), which may or may not have direct and clear relationships to effort. While no statements about effort appear in the conclusions section, they do appear in the Discussion and Abstract. Unless the authors have evidence that steer rate is strongly correlated to effort/steering torque (please provide a reference), statements about effort should be excluded from the manuscript.

Response: We appreciate the valid point raised. We acknowledge that kinematics, such as steer, roll, and lean rates, may not directly and clearly indicate rider effort without further evidence or correlations to steering torque or related measures. To ensure accuracy and avoid making unsupported claims, we revised the manuscript to remove references to steering effort and changed it to required steering actions or simply steer rate.

**Comment 5:** The language and tone used in the manuscript are appropriate.

**Comment 6:** The conclusions are supported by the results presented in the manuscript.

**Comment 7:** The limitations of the study are discussed.

**Comment 8:** The implications of the study are discussed.

**Comment 9:** The authors suggest future directions for research.

## Additional comments

**Comment 1:** Overall, I enjoyed reading this manuscript. However, I feel the impact and value of the results are limited by the limited description of the balance assist algorithm.

Response: We apologize for the limited description of the balance assist algorithm. We have now provided additional details in the manuscript regarding the design and implementation of the controller. This includes information on the basic concept, design, and tuning process, which will help enhance the understanding of the balance assist algorithm.

**Comment 2:**Design and interpretation of the controller: If I interpret this correctly, the controller is using a bicycle model that uses the benchmark bicycle parameters, correct? Is the controller adapted in any way to account for different bicycle geometries or riders?

Response: Yes, indeed our controller operates on benchmark bicycle parameters. To address your concern about different bicycle geometries and riders, we believe that as long as the specific parameters of the bicycle are known, the characteristic matrix A can be calculated effectively and we can identify the self-stable and unstable regions and calculate the required steer torque required to shift the eigenvalues to the stable region.

While we ran the simulation for the benchmark bicycle with a rigid rider, we took some additional measures to address bicycle and rider variability. We have first kept the M, C, and K matrices constant and tested different gains, then we kept the gain constant and slightly changed the M matrices to see if the previously stable region becomes unstable with slightly different parameters. The gain we chose showed a robust result for these changes. Additionally prior to the main experiment, we conducted a series of pilot tests to assess the slowest speed a rider could ride while maintain an upright position

without their hands on the handlebars. The results indicated that balance could be maintained at speeds as slow as 1.4 m/s. These findings were promising and gave us confidence in proceeding with the experiment with controller based on the benchmark bicycle. Therefore, we believe that the controller's design and interpretation take into account variability among riders and bicycles to a sufficient extent.

**Comment 3:** Figure S4: Can you provide some additional details regarding the 'Current command from Teensy to Steering motor' (what is it and how does it relate to the experiment)? Are these data from a disturbed trial?

Response: We apologize for the lack of clarity regarding the 'Current command from Teensy to Steering motor' in Figure S4. This refers to the current command signal sent from the Teensy microcontroller to the steering motor to control the steering action. We have now made it clear in all figures by converting it to torque, but removed it from this figure as the focus of this figure is to demonstrate how we segmented the data, regardless of the condition.

**Comment 4:** The main revisions I would like to see are as follows (as discussed above): a. Provide additional details about the controller for balance assist. b. Provide additional details for the citations that I was unable to locate. c. Do not make claims about effort unless you can provide convincing evidence that steer rate is highly correlated to steering torque. d. Provide additional details about the experimental design (randomization of trials, cadence, rider posture).

Response: We appreciate the feedback and suggestions for revisions. We have made the necessary updates to address each of these points in the manuscript.

The authors would like to thank you for your valuable feedback and suggestions. We believe that the revisions we have made based on your comments have significantly improved the quality and clarity of our paper. We look forward to hearing your response and hope that the revised version meets your expectations.

# Reviewer 2

## Additional comments

This paper evaluates the effectiveness of a balance assist system in older and young adults while cycling a straight line (with or without a dual task and perturbations). This is a well-written paper. I have several minor comments/suggestions that might improve the readability of the paper.

Response: Thank you for your positive feedback. We appreciate your suggestions for improving the readability of the paper.

**Comment 1:** Methods L 88 – 95: You mention that you manually tuned the gains, but it is unclear how this was done? It is understood that you select gains that result in a locally stable controller (negative real parts eigenvalues), but how can you minimize required steering torque? Do you mean minimize the feedback gains (and hence minimize required steering torque)? You can maybe make this more explicit for the reader.

Response: Thank you for your insightful feedback. We have revised the methods section to provide clarity on how we manually tuned the gains to ensure stability and minimize steering torque. Additionally, we have included brief details about pilot tests with riders, demonstrating that the chosen gains resulted in smooth, responsive, yet non-aggressive control. We believe these changes will improve the manuscript's clarity and depth (lines 84-127).

**Comment 2:** Methods L95: Slightly different results are to be expected in real scenarios. I assume

that you only expect slightly different results when the subjects are indeed moving on the bicycle but do not control the steering angle. I guess that you expect a large difference when the subject is also controlling the steering angle. You might want to clarify this to the reader...

Response: Thank you for your keen observation and constructive feedback. In response, we have made modifications to our methods section for better clarity. The updated section can be found in lines 120-126.

**Comment 3:** L77-95: Maybe make it more explicit that you are talking about local stability of the system (around the state of upright position of the bike with zero velocities) and not stability in general.

Response: Thank you for your insightful feedback. Your point about the clarity regarding the term 'stability' is well taken. In our manuscript, we indeed focused on the dynamics of the bicycle linearized about the upright, zero-steer, and constant forward speed configuration. Our usage of the term 'stability' was in the context of this particular state. We understand that without explicit reiteration of this point in this section, the context may have been obscured. The section has now been updated for further clarification and reads as:

'Our control objective was to stabilize the system at all forward speeds, specifically within the linearized dynamics about an upright position, with a zero-steer angle, and a constant forward speed.'

**Comment 4:** Figure 1: It might be easier for the reader if you add legends for different colors (eigenvalue 1, 2, 3). The reader might also like a black line for the maximal value of the real parts of eigenvalues (you use this to determine if the system is locally stable) and maybe highlight locally unstable from 1 - +/- 4.5 m/s in the controller off condition.

Response: Thank you for your feedback. We have made updates to Figure 1 to enhance its clarity and improve understanding. Legends have been added to differentiate between the different eigenvalues (1, 2, 3). We have also visually highlighted the unstable region and separated the real parts of the eigenvalues for low and high forward speeds. Additionally, the imaginary parts of the eigenvalues are now displayed in a separate subplot. The caption of the figure has been revised to provide a clearer interpretation of the figure's content.

**Comment 5:** L103-110: You did not explain how you measured the roll angle and rate at this point (I assume using an IMU on the frame, is this correct?).

Response: Thank you for your comment. We have clarified our measurement methods in the manuscript to clarify this (lines 140-141).

**Comment 6:** L111: A simple equation for feedback laws + reporting feedback gains would be nice in the main manuscript.

Response: Thank you for your valuable suggestion. We have now extensively modified the methods section and incorporated the equation for feedback law and included the feedback gains in the main manuscript (lines 84-127).

**Comment 7:** Figure 2: You might want to illustrate the roll axis in this figure? (This might help some readers without a mechanics background).

Response: Thank you for your suggestion. We have now incorporated bicycle axes in Figure 2 of the manuscript to better illustrate the roll and steering axes.

**Comment 8:** Figure 3: I think that you can improve this figure. Given the relatively small number of subjects, I think that you have to show the individual data points or visualize the distribution (using, for example, a violin plot). As the balance assist on-off is a within-subjects variable (repeated

measures), you might also want to connect the data points per subject with a line as you do the statistics on the difference between assist on-off for each subject.

Response: We agree with your suggestion to improve Figure 3 and 4. We have modified the figure to include individual data points and we have connected the data points per subject with a line to enable visualization of the balance assist on-off conditions as a within-subjects variable.

**Comment 9:** L364-366: You might want to rewrite this sentence. I understand that the effect of the balance assist system, in general, is independent of the test location as this is a within-subjects variable. However, this sentence now suggests that the conclusions about age are also independent of the test location, which is not the case. The revised sentence now accurately reflects this distinction.

Response: Thank you for pointing out the need for clarification in that sentence. We have rewritten it to accurately reflect the distinction you mentioned (lines 408-419)

The authors would like to thank you for your valuable feedback and suggestions. We believe that the revisions we have made based on your comments have significantly improved the quality and clarity of our paper. We look forward to hearing your response and hope that the revised version meets your expectations.

Sincerely,

Leila Alizadehsaravi